# A Multi-Task Fusion Strategy-Based Decision-Making and Planning Method for Autonomous Driving Vehicles

**DOI:** 10.3390/s23167021

**Published:** 2023-08-08

**Authors:** Weiguo Liu, Zhiyu Xiang, Han Fang, Ke Huo, Zixu Wang

**Affiliations:** 1Information Science & Electronic Engineering, Zhejiang University, Hangzhou 310027, China; xiangzy@zju.edu.cn; 2National Innovation Center of Intelligent and Connected Vehicles, Beijing 100176, China; huoke@china-icv.cn (K.H.); wangzixu@china-icv.cn (Z.W.); 3Wuhan Lotus Technology Co., Ltd., Wuhan 430000, China; fanghan3@lotuscars.com.cn

**Keywords:** deep reinforcement learning, decision-making planning, multi-task fusion, DDPG, simulation platform, end-to-end, VTD

## Abstract

The autonomous driving technology based on deep reinforcement learning (DRL) has been confirmed as one of the most cutting-edge research fields worldwide. The agent is enabled to achieve the goal of making independent decisions by interacting with the environment and learning driving strategies based on the feedback from the environment. This technology has been widely used in end-to-end driving tasks. However, this field faces several challenges. First, developing real vehicles is expensive, time-consuming, and risky. To further expedite the testing, verification, and iteration of end-to-end deep reinforcement learning algorithms, a joint simulation development and validation platform was designed and implemented in this study based on VTD–CarSim and the Tensorflow deep learning framework, and research work was conducted based on this platform. Second, sparse reward signals can cause problems (e.g., a low-sample learning rate). It is imperative for the agent to be capable of navigating in an unfamiliar environment and driving safely under a wide variety of weather or lighting conditions. To address the problem of poor generalization ability of the agent to unknown scenarios, a deep deterministic policy gradient (DDPG) decision-making and planning method was proposed in this study in accordance with a multi-task fusion strategy. The main task based on DRL decision-making planning and the auxiliary task based on image semantic segmentation were cross-fused, and part of the network was shared with the main task to reduce the possibility of model overfitting and improve the generalization ability. As indicated by the experimental results, first, the joint simulation development and validation platform built in this study exhibited prominent versatility. Users were enabled to easily substitute any default module with customized algorithms and verify the effectiveness of new functions in enhancing overall performance using other default modules of the platform. Second, the deep reinforcement learning strategy based on multi-task fusion proposed in this study was competitive. Its performance was better than other DRL algorithms in certain tasks, which improved the generalization ability of the vehicle decision-making planning algorithm.

## 1. Introduction

At present, autonomous driving technology can be divided into two major categories [1]. As depicted in Figure 1, one is a modular autonomous driving system based on rule-based construction, and the other is an end-to-end autonomous driving system based on neural networks. Rule-based systems make autonomous driving decisions by constructing a rule library. By statistically analyzing the situations that the autonomous vehicle may encounter and then establishing rules between vehicle driving states and corresponding strategies, the above-mentioned rules are used to control the vehicle [2]. However, the real driving environment is complex and constantly changing, and it is impractical to construct a complete rule library. Moreover, when the autonomous vehicle encounters situations that are not included in the rule library, the probability of accidents is increased. Accordingly, rule-based autonomous driving systems cannot conform to needs. Compared with rule-based systems, deep learning-based systems do not require humans to develop a rule library but only to adopt an end-to-end neural network to control the vehicle such that costs can be significantly lowered. However, the ability of the algorithm to abstract autonomous driving scenes is limited, and it cannot cope with complex driving scenarios. In 2017, Xu et al. [3] from the University of Berkeley proposed a method for training autonomous vehicles end-to-end using large-scale video datasets. The method uses the FCN-LSTM architecture to learn from large-scale video datasets and can predict the future vehicle motion distribution. Moreover, the method combines scene segmentation tasks to improve algorithm performance. However, such deep learning-based methods require huge datasets, and the resulting autonomous vehicles have limited exploration capabilities.

Imitation learning refers to a method of learning strategies through a large amount of human expert data. In 2011, Ross et al. [4] proposed a data aggregation algorithm (Dataset Aggregation (DAgger)). To be specific, this method employs human driving data to train an initial policy. Subsequently, it implements this policy to interact with the environment to produce new states, then expert experience is adopted to label the actions of the new states, and then it repeats the process of training the policy network. The DAgger algorithm can learn human driving skills, but it cannot learn strategies beyond expert experience. Although the above DRL and imitation learning methods have numerous advantages, they still use gradient descent to optimize parameters. However, practical scenarios may be complex and gradient-free. Thus, conventional algorithms based on gradient descent for parameter optimization cannot be used when the gradient cannot be addressed in the autonomous driving environment.

Compared with deep learning, deep reinforcement learning does not require artificially labeled data, while it exhibits excellent exploration ability. In 2017, Hyunmin et al. [5] proposed to apply the Deep Q-learning Network (DQN) algorithm to research on automatic braking systems of cars. As indicated by the result of nearly 70,000 experiments performed in a simulation environment, the algorithm is capable of learning the skill of automatic braking. Peter Wolf et al. [6] proposed a DQN-based autonomous driving algorithm in 2017. The above-mentioned method is only dependent on the image data obtained with the on-board camera as the input of DQN while discretizing the steering wheel action. The self-driving car is directly trained end-to-end in accordance with the Q value determined by the DQN algorithm for the turning action. In addition, this car achieves the identical level of performance as human driving in a simulation environment. However, the above-described DRL methods based on trial-and-error training at early stages require a long time for training, conform to poor early-stage strategies, exhibit a slow learning speed, and cannot learn skills beyond human ability.

Later DRL algorithms are primarily optimized based on the DQN algorithm and the policy iteration algorithm, comprising the Double DQN (DDQN) algorithm [7], DDPG algorithm [8], Advantage Actor–Critic (A2C) algorithm [9], Proximal Policy Optimization (PPO) algorithm [10], Twin Delayed Deep Deterministic Policy Gradient (TD3) algorithm [11], and so forth, which have achieved favorable results in games.

Sample efficiency is considered one of the most significant challenges facing reinforcement learning. The reward signal is sparse, delayed, or noisy in numerous practical scenarios, resulting in a low learning efficiency since most collected experiences do not generate learning signals. Auxiliary tasks can lead to an improvement in data efficiency by forcing the agent to learn auxiliary predictions and control objectives beyond maximizing rewards such that better representations can be produced. Banafsheh Rafiee et al. [12] investigated the usefulness of auxiliary tasks in reinforcement learning. The research was aimed at gaining insights into how auxiliary tasks help in learning the main task. Sun et al. [13] proposed an auxiliary-task-based reinforcement learning framework with the aim of addressing the problem of heterogeneous observations from multiple sources in autonomous driving. The proposed framework enables the network to represent scene-associated features without the need for additional information, supporting the policy network to make appropriate decisions in a wide variety of scenarios. For instance, with two auxiliary tasks (i.e., depth map prediction and loop closure detection) combined in navigation, the agent is enabled to learn to navigate in complex 3D mazes based on raw sensory inputs. On that basis, the agent is capable of achieving close to human-level performance even in scenarios in which the target location frequently varies [14]. In the 3D map visual navigation task, the reward prediction auxiliary task has been introduced to the Asynchronous Advantage Actor–Critic (A3C) algorithm such that the algorithm outperforms previous state-of-the-art techniques on Atari [15]. In autonomous driving, classic auxiliary tasks comprise depth estimation [16], semantic segmentation [17], or optical flow. The above-mentioned methods require additional auxiliary information for computation. The focus of this study was placed on data augmentation and semantic segmentation, and a method to incorporate this information into data augmentation, pre-training, and auxiliary tasks during training was investigated. Semantic segmentation encompasses almost all the elements required for urban driving (e.g., distinguishing roads and sidewalks and detecting cars and pedestrians). It is noteworthy that an agent should exhibit the capability of performing the above-described tasks even in previously unseen environments [18,19].

In brief, although the above autonomous driving research has made certain achievements, there are still several shortcomings. The first is the insufficient acquisition of driving environment information. If only the camera image serves as the input of the system, only obstacles around can be detected, and accurate position information of obstacles is difficult to obtain, especially in extreme weather such as rain and snow, which will produce noise in the camera image and greatly affect the model’s ability to acquire the surrounding environment. The second is the insufficient utilization of global path and other navigation information. The task of autonomous driving is generally using a means of transportation when people want to go from one place to another without requiring manual intervention, and the means of transportation can independently complete the driving task. However, many DRL end-to-end algorithms have not considered navigation information, and their autonomous driving tasks accurately say that they allow vehicles to roam aimlessly on the road. The above-mentioned problems are urgent problems to be solved in the field of autonomous driving and also were the focus of this study. In the present section, the classic techniques used in computer vision to reduce overfitting are first briefly outlined, and then the introduction of information to improve performance and generalization is investigated. As indicated by the results, this framework can effectively expedite the learning process and achieve better performance. The contributions of this study are as follows:A DDPG decision-making and planning strategy method was proposed for multi-task fusion based on deep reinforcement learning that integrated three dynamic information types (i.e., an image semantic segmentation task to encode results, global path-planning results, and the real-time vehicle state to achieve the behavior decision-making task of the intelligent vehicle). The problem of a low-sample learning rate arising from sparse reward signals was effectively resolved.A co-simulation testing and verification platform based on VTD + CarSim + Tensorflow + Simulink was designed and implemented. The simulation platform was adopted to test and verify the decision-making and planning system, and the experimental results were analyzed.In accordance with the motion characteristics of the ego vehicle and obstacles in dynamic scenarios, the reward function of the trajectory was re-optimized.

## 2. Materials and Methods

### 2.1. Deep Deterministic Policy Gradient

DDPG refers to a deterministic policy gradient method that employs deep neural networks to approximate the deterministic policy and value function. Deterministic policy gradient methods refer to policies for which the action is unique for the identical policy and state. Since the actions of a deterministic policy are fixed, the solution of the policy gradient does not require sampling and integrating over the action space such that fewer samples are required. However, since the trajectories generated by a deterministic policy are fixed, the agent cannot access other states or learn new behaviors.

To address the above-mentioned problem, an off-policy learning method should be developed. Off-policy learning refers to using different policies for action and evaluation. In the action policy, a stochastic policy is employed to ensure sufficient exploration, while the evaluation policy is a deterministic policy, i.e., $a_{t}=\pi_{\theta }\left(s_{t}\right)$.

The entire framework of deterministic policy learning uses the Actor–Critic method. The heterogeneous deterministic policy gradient calculation method is:(1)$$\nabla_{\theta }J_{\beta }\left(\mu_{\theta }\right)=E_{s\sim \rho^{\beta }}\left[{\nabla_{\theta }\mu_{\theta }\left(s\right)\nabla_{a}Q^{\mu }\left(s,a\right)|}_{a=\mu_{\theta }\left(s\right)}\right]$$
where β denotes the sampling policy, ρ represents the state distribution, $\mu_{\theta }\left(s\right)$ is the deterministic policy, and $Q^{\mu }\left(s,a\right)$ expresses the action-value function. The update process of the deterministic policy using the Actor–Critic algorithm is written as:(2)$$\delta_{t}=r_{t}+\gamma Q^{a}\left(s_{t+1},\mu_{\theta }\left(s_{t+1}\right)\right)-Q^{a}\left(s_{t},a_{t}\right)$$
(3)$$\omega_{t+1}=\omega_{t}+\alpha_{\omega }\delta_{t}\nabla_{\omega }Q^{\omega }\left(s_{t},a_{t}\right)$$
(4)$$\theta_{t+1}=\theta_{t}+{\alpha_{\theta }\nabla_{\theta }\mu_{\theta }\left(s\right)\nabla_{a}Q^{\omega }\left(s_{t},a_{t}\right)|}_{a=\mu_{\theta }\left(s\right)}$$
where Equation (2) is the equation for the value function error term, $r_{t}$ represents the reward at the previous time step, $Q^{\omega }\left(s_{t+1},\mu_{\theta }\left(s_{t+1}\right)\right)$ represents the current *Q* value under the policy ω, and $Q^{\omega }\left(s_{t},a_{t}\right)$ represents the *Q* value at the previous time step. Equations (3) and (4) represent the methods for updating the value function parameters and the policy parameters using the value function approximation, respectively, where $\alpha_{\omega }$ and $\alpha_{\theta }$ are the learning rates for the value function and policy function, respectively, and $\mu_{\theta }\left(s\right)$ is the deterministic policy. Using the above-described method, the parameters of the deterministic policy can be searched and updated, and an optimized action output can be determined for the current state. However, unstable results are generally obtained using deep neural networks for function approximation since deep neural network training often assumes that the data follows independent and identically distributed distributions. Nevertheless, RL training data refers to a sequential time series, and the Markov property of the data makes it unable to conform to the condition of being independent and identically distributed. To break the data correlation, DDPG employs two techniques (i.e., experience replay and an independent target network). Experience replay was proposed by Lin in 1995 for the first time [20], mainly to address the correlation and non-stationarity of training data. The specific method aims at storing the data generated by the agent–environment interaction in an independent sample pool and then using the uniform random sampling method to extract data from the sample pool to train the neural network. The loss function for training the AC model is written as:(5)$$L\left(\theta^{Q}\right)=E_{s_{t}\sim \rho^{\beta },a_{t}\sim \beta ,r_{t}\sim E}\left[Q\left(s_{t},a_{t}|\theta^{Q}\right)-y_{t}{)}^{2}\right]$$
(6)$$y_{t}=r\left(s_{t},a_{t}\right)+\gamma Q\left(s_{t+1},\mu \left(s_{t+1}\right)|\theta^{Q}\right)$$
where $y_{t}$ is the practical reward value at the current time step, and $Q\left(s_{t},a_{t}|\theta^{Q}\right)$ and $Q\left(s_{t+1},\mu \left(s_{t+1}\right)|\theta^{Q}\right)$ are the *Q* values at the previous and current time steps, respectively. The independent target network technique adopts the main network for single-step learning and iterative updating. After a certain number of iterations, the parameters of the main network are assigned to the independent target network. In terms of DDPG, the independent target network approximates the parameters of the main network by a small amount of variation each time. The update equation for DDPG using the independent target network is written as follows:(7)$$\delta_{t}=r_{t}+\gamma Q^{a^{\prime }}(s_{t+1},\mu_{\theta^{\prime }}\left(s_{t+1}\right)\;)-Q^{a}\left(s_{t},a_{t}\right)$$
(8)$$\theta^{\prime }=\tau \theta +\left(1-\tau \right)\theta^{\prime }$$
(9)$$\omega^{\prime }=\tau \omega +\left(1-\tau \right)\omega^{\prime }$$

### 2.2. The DRL System Workflow

The optimization goal of this study was to directly control the driving actions of a vehicle (e.g., acceleration, deceleration, and steering) based on the vehicle’s state and environmental information (e.g., vehicle speed, camera images, and waypoints) in an urban road operating environment and to reach the destination smoothly and rapidly. The input was the image captured by a camera installed on the vehicle, and the control commands were the steering wheel angle, throttle pedal opening, and brake pedal opening. Accordingly, the problem first was required to be transformed into a reinforcement learning problem. Figure 2 illustrates the overall system structure. The actor network output actions, the critic network estimated action Q-values, the experience replay pool stored exploration data, the VTD–CarSim was the vehicle-running interactive environment, and the reward function output the action rewards. The performance in the environment was enhanced through continuous interaction between the agent and the environment based on the reward function. The entire system operation fell into two parts (i.e., environment exploration and network training).

In this study, the VTD 2020 software served as a simulation scenario rendering tool that could provide users with a description of the vehicle environment (e.g., whether it was in the lane and the position of nearby obstacles) while providing images from different perspectives and switching between vehicles. The RGBD forward camera model was adopted to provide observation input, which could be represented by a tensor of RGBD @224 × 224 × 4. RGB and depth images were employed together since they are important sensor sources and provide complementary information. The depth image provided relative spatial information for other road participants and obstacles, whereas the RGB image was effective in providing road semantic information. Second, the CarSim 2020.1 software served as a dynamic model that could provide a description of the vehicle state (e.g., engine speed and gear position). Moreover, the controller was capable of performing typical vehicle control operations (e.g., engaging the clutch, gear shifting, acceleration, braking, and steering). All the above-mentioned simulation data were accessed through a simulation interface to conform to the needs of algorithm training.

As shown in Figure 2, the actor network had two input modules (the driving environment information feature module and the navigation status information module). It used tensor concatenation to combine the two feature tensors and then passed through two hidden layers with 256 neurons each; the output layer had three neurons that output the vehicle’s actions, including controlling the steering wheel angle, throttling, and braking.

The critic network, also shown in Figure 2, consisted of an input driving environment information feature module, a navigation status information module, and an action module. It used tensor concatenation fusion to combine the three feature tensors and then passed through two hidden layers with 256 neurons each; the output layer had one neuron that output Q-values. The action input module consisted of two fully connected layers with 128 neurons each.

### 2.3. The Neural Network Architectures

Figure 3 shows a reinforcement learning architecture with semantic segmentation as an auxiliary task, which has the capability of multi-modal sensor fusion for end-to-end autonomous driving and semantic segmentation. It can be implicitly divided into three parts: the multi-modal sensor fusion encoder-decoder, navigation state information extraction module, and waypoint feature extraction module. The RGB image of W × H × 3 and depth image of W × H × 1 (with width W and height H (224 × 224)) were first concatenated by channels to form an RGBD structure and then input into the multi-modal sensor fusion encoder, conforming to the ResNet structure [21]. The output of the encoder was expressed as a 7×7×2048 feature map, which was then connected to the semantic segmentation decoder consisting of five deconvolution layers, with the last deconvolution layer using the softmax activation function. Except for the last layer, a batch normalization layer was followed by a ReLU non-linear function after the respective deconvolution layer. The number of filters in the five deconvolution layers was 512, 128, 64, 16, and 5, respectively, and the kernel size of the respective filter in the deconvolution layers was set to 3 with a stride of 4, 2, 2, 2, and 1, respectively. Lastly, the decoder output the category of the respective pixel in the original image to represent its understanding of the driving scene. The decoder output $S_{t}\in {\left[0,1\right]}^{\left(W\times H\times C\right)}$ as the number of categories to be segmented. For a given pixel in the original image, the output was the probability that the corresponding pixel belonged to different categories. The above-described five categories were the same as those used in the previous semantic segmentation training: road, lane, vehicle, sidewalk, and background. The feature map of 7×7×2048 was globally average-pooled to produce a latent feature vector. Next, the vector was merged using tensor concatenation to form the driving environment information feature.

In this study, a mixed state space was chosen as the input of the neural network, shown as:(10)$$s_{t}=\left\{s_{t}^{RGBD},s_{t}^{speed},s_{t}^{angle},s_{t}^{waypoint}\right\}$$
where $s_{t}^{speed}$ represents speed state information, $s_{t}^{RGBD}$ represents input image information, $s_{t}^{angle}$ represents steering angle information, and $s_{t}^{waypoint}$ represents the waypoint feature. To be specific, the agent read RGBD images at each simulation step. Additionally, extra motion information was used to achieve better control performance. Speed data was extracted from CarSim to generate the speed vector $s_{t}^{speed}$, which included longitudinal speed u, lateral speed v, engine speed $v^{eng}$, and four-wheel speed $v_{RF}^{whl},v_{RR}^{whl},v_{LF}^{whl},v_{LR}^{whl}$, as well as the steering angle.
(11)$$s_{t}^{speed}={\left[u,v,v^{eng},v_{RF}^{whl},v_{RR}^{whl},v_{LF}^{whl},v_{LR}^{whl}\right]}^{T}$$

We investigated the effect of adding semantic segmentation as an auxiliary task. An auxiliary task refers to training a task different from the main task simultaneously to improve its performance. In this example, semantic segmentation was trained simultaneously with driving decision making. One challenge was to combine the two types of learning together rather than having one type dominate. The general loss was a weighted sum of different losses, shown as:(12)$$l=\lambda_{rl}l_{rl}+\lambda_{aux}l_{aux}$$
where $l_{rl}$ denotes the DDPG loss, $l_{aux}$ represents the semantic cross-entropy loss, and $\lambda_{rl}$ and $\lambda_{aux}$ express the weights for the respective loss. This simple combination of losses was first examined, and the weight of the auxiliary task should be tuned.

The $l_{aux}$ is used to measure examine the difference between a semantic model’s predicted output and the GT. In general, the smaller the value of the loss function, the better the robustness of the model will be. In addition, the cost function represents the average of the loss function over the entire sample set. Its equation is written as:(13)$$l_{aux}=-\sum_{i}p(x_{i})\log(q(x_{i}))$$
where *p* denotes the GT and *q* expresses the predicted output.

A more effective adaptive auxiliary task weighting method described in reference [22] was employed to perform automatic loss balancing in the multi-task learning. It was examined, and effective results were obtained by combining the main task and the auxiliary task. At time step t, the loss comprised the network weight $\theta_{t}$ and the auxiliary task weight w. The loss function is expressed in Equation (14):(14)$$l\left(\theta_{t}\right)=w\times l_{aux}\left(\theta_{t}\right)+l_{main}\left(\theta_{t}\right)$$
where the gradient update of w is proportional to the product of “the gradient of $l_{main}\left(\theta_{t}\right)$ with respect to the parameter and the dot product of the gradient of $l_{aux}\left(\theta_{t}\right)$ with respect to the parameter and the auxiliary task”. This loss was used for all subsequent experiments.

In the training process, the algorithm enhanced the performance of the main task by automatically updating the class weights in the auxiliary tasks. The updated weights affected by Equation (14) affected the learning process. To be specific, if certain class labels were detrimental to the learning of the main task or if the main task had already acquired all relevant information from those labels, their weights would approach zero, with the aim of preventing them from interfering with the learning of the main task. In addition, if certain labels in the auxiliary tasks were conducive to the learning of the main task, their weights would indicate their significance in the learning process while affecting the learning process through backward propagation of the loss.

### 2.4. The Simulation Environment Setup

Nowadays, surreal virtual testing turns out to be a vital concept in the construction of safe AV technology. The use of photo-realistic simulations (virtual development and validation testing) and appropriate driving scene design take on critical significance in building safe and reliable AVs. For DRL algorithms, the complexity of urban environments requires the above-mentioned algorithms to be examined in countless environmental and traffic scenarios. The above-described problem leads to a doubling of costs and development time when physical methods are applied. Thus, simulation software such as VTD, which is currently one of the most powerful and promising simulators for developing and testing AV technology, is used.

VTD refers to an extensively used open simulation platform worldwide for virtual testing and validation of ADAS and autonomous driving vehicles. First, it allows for numerous tests to be performed as required, avoiding putting lives or goods at risk and reducing development costs and implementation time. It is unlikely to conduct projects of this nature (training DRL algorithms for AV navigation purposes in any complex scenario) directly in a real environment since this will pose a risk to the ego and its surrounding environment, especially at the start due to the randomness of the first action taken by the algorithm. Second, since there are numerous datasets regarding the vehicle perception layer (e.g., segmentation or object detection and tracking), to verify the effectiveness of control algorithms, they should be compared with the ideal route that a vehicle should follow. For the control layer, VTD provides users with the practical mileage of the vehicle as well as the true situation of the route, making it easier to evaluate the performance of the proposal.

In this study, a joint simulation platform of VTD + CarSim + Simulink + Tensorflow was designed and developed, as presented in Figure 4. Co-simulation of VTD and CarSim provided a testing environment for the examined algorithm in which VTD simulated the environment and sensors and CarSim simulated the dynamic vehicle. The synchronization between the two was achieved as follows: CarSim sent the (x, y) information of the four tire–road contact points to VTD. The odrGateway module in VTD checked the OpenDRIVE map while returning the height information (z) of the four tire contact points; this was adopted by CarSim for dynamic calculations. Simultaneously, CarSim sent the vehicle’s state information (e.g., position and orientation) to VTD, allowing VTD to perform dynamic visualization and update the sensor information.

In the above-mentioned solution, VTD and CarSim achieved information exchange through Simulink. VTD utilized the UDP send/receive module in Simulink for communication, while CarSim provided a communication method with Simulink through the S-function. The TensorFlow-based algorithm was capable of developing communication interfaces in Simulink based on specific requirements to obtain the vehicle state and sensor information and send control commands.

Interfaces were developed through Socket nodes that were connected to VTD to receive information. The VTD software provided the necessary components (e.g., traffic objects, road layouts, and infrastructure) for conducting physical-based and sensor-based testing while performing scene-based AV testing. Components that required low-frequency control (e.g., the mobility of traffic vehicles and pedestrians, the state of traffic lights, and the emergence of new traffic participants) were processed by the Simulation Controller Protocol (SCP). High-frequency data exchange (e.g., traffic object and sensor information between VTD and external libraries) was performed using a runtime data bus (RDB). The interface developed in Tensorflow to interact with VTD was achieved through shared memory interfaces and network interfaces.

Shared memory (SHM) interfaces were adopted to exchange high-frequency sensor data. They were implemented through RDB in VTD. RDB is a proprietary binary communication protocol used by VTD that exports simulation data in its own packet format. Figure 5 displays the flowchart of the shared memory interface with the camera and LI-DAR sensors. Shared memory (SHM) nodes were developed to extract the camera and LIDAR sensor data, which were written into two different memory segments in RDB format. The RDB packet of the sensor comprised the simulation time, frame number, packet ID, pixel format, message size, and data array. Subsequently, the sensor RDB packet was read by client nodes connected to a specific SHM segment, which parsed the packet and formatted it into a sensor message while publishing it to the corresponding client topic. The above-mentioned sensor data was read in the respective simulation step, and the frequency of the camera could reach up to 30 fps.

As depicted in Figure 6, the simulation process primarily covered the following steps. First, a virtual scene was set in the simulation software. Subsequently, the simulation was initiated, and the states were synchronized. Third, CarSim and VTD interacted with the first frame data to align the initial points. Fourth, VTD provided feedback on the current position and environmental information of the vehicle to CarSim. Lastly, CarSim simulated a wide variety of modules of the vehicle and calculated the vehicle’s position and attitude information based on the current position and environmental information. The above-described steps were repeated until the test sequence was completed, and then the stop button was triggered to control VTD to stop running.

CarSim, a vehicle dynamics simulation tool, is capable of simulating the dynamic response of a vehicle through convenient parameter settings and serving as the controlled object for algorithms in the system. CarSim combines vehicle dynamics modeling methods with multi-body system dynamics modeling methods to abstractly simplify the vehicle system. In general, it comprises 10 rigid body systems (i.e., the vehicle body, four sprung masses, four rotating wheels, and the engine crankshaft). The drivetrain and brake system drive and brake the wheels, while the steering system characteristics and the suspension’s K&C properties control the wheel steering, determining tire motion. Tire forces are then determined using the tire model. Table 1 lists the dynamic simulation parameters of the vehicle model in CarSim.

## 3. Results

### 3.1. Design of Reward

In reinforcement learning algorithms, designing a well-defined task reward takes on critical significance in guiding the learning process. A simple and direct reward for autonomous driving can act as the distance a car travels before crashing. However, this type of reward signal lacks information and cannot collect multi-dimensional information to guide an intelligent agent’s learning. Accordingly, a specific task reward function is defined to guide the vehicle to stay in its lane and perform lane-changing operations when possible. Here, specially designed reward signals were introduced, mainly designed based on the three aspects of traffic efficiency, safety, and comfort. The final reward comprised the following five parts, as presented in Figure 7.

1.In the experiment, it was expected that the direction of the vehicle’s speed was consistent with the direction of the road. Thus, the speed vector in the direction of the road was rewarded, and the speed vector deviating from the road direction was punished. To be specific, θ represented the angle of the velocity vector direction deviation from the road direction.
(15)$$r_{1}=cos|\frac{2}{3}\theta |-\sqrt{3}cos|\frac{2}{3}\theta |$$

2.Since the aim was to make the vehicle stay in the center of the lane while driving, any lateral deviation from the center of the lane was penalized. To be specific, *d* represented the current distance to the center of the lane.
(16)$$r_{2}=-|d|$$

3.A large negative punishment had to be given when the vehicle drove out of the road boundary. If the vehicle entered this state, the round would end and a new round would start.
(17)$$r_{3}=-1\left\{Out\;of\;Boundary\right\}$$

4.The reward encouraged the vehicle to achieve a greater speed but not exceed the target speed of 35 m/s.
(18)$$r_{4}=35-|35-v|$$

5.With the aim of changing lanes, if there were vehicles ahead within a range of 100 m, the vehicle had to be encouraged to overtake. To be specific, x represented the distance to the preceding vehicle on the identical lane. If no vehicles were identified, the default value of x was 100.
(19)$$r_{5}=-max\left(\left(0,100\right]-x\right)$$

6.Regarding the amplitude of the vehicle’s steering angle, this reward parameter acted as a punishment for the vehicle making large steering wheel turns. It was expected that the agent could make the vehicle’s steering wheel smoother when performing behaviors such as turning corners and avoiding obstacles such that the amplitude of the steering wheel angle δ was considered.

(20)$$r_{6}=-k_{s}\times max\left(\left(|\delta |-0.2\right),0\right)$$
where $k_{s}$ denoted the penalty coefficient for the vehicle’s steering angle, and the output range of the steering wheel angle was expressed as [−1, 1]. $r_{6}$ was the reward item for the vehicle’s direction steering angle.

The overall reward function was a linear combination of the above items with specified weights:(21)$$R=\sum_{i=1}^{6}w_{i}r_{i}$$
where the reward was first normalized to the range (0, 1). Subsequently, a good weight vector w was searched and generated. More powerful models will test different weighting coefficients to find the best combination. A future possible improvement refers to using inverse reinforcement learning to automatically extract the parameters of the reward function.

### 3.2. Evaluation and Analysis

In the present section, a method for setting the experimental environment and implementing the algorithm is demonstrated. In addition, different methods applied in the experiment will be quantitatively evaluated and compared.

#### 3.2.1. Experiment Setup

Due to the training process of the deep reinforcement learning model involved in the algorithm proposed in this study, a large amount of data was generated during the interaction training process between the agent and the environment that needed to be stored in the memory space to establish a sample database for the interaction between the agent and the environment, thereby enhancing the learning ability of the agent. Accordingly, the decision-making and planning method proposed in this study relied on computers with strong CPU and GPU computing capabilities and a large memory space. Table 2 lists the practical software and hardware environment used in the experiment.

TensorFlow is an API that can be used to quickly build advanced neural networks using the Python programming language. TensorFlow is characterized by its user-friendliness, modularity, ease of extension, and Python implementation. The above-mentioned flexible architecture is capable of deploying computational tasks into one or more CPUs or GPUs on desktop devices, servers, or mobile devices through porting.

The experiments in this section fell into three stages. The difficulty of the respective stage varied, and the overall difficulty trend was gradually increased. Moreover, the respective stage comprised two different weather environments, i.e., sunny and rainy days. The task of the respective stage aimed at driving the vehicle to the destination. The focus of the task was placed on testing the performance of the decision-making and planning task of the vehicle under straight-line driving, turning, and dynamic traffic, so the effect of traffic lights and lane yellow lines was not considered during the training and testing process. The test map applied in this experiment was Germany_2018 in VTD, i.e., a city map with a circular eight-lane road, including overpasses, roundabouts, slopes, three-way intersections, and other scenes. The specific map is shown in Figure 8.

Task settingThe specific task settings for the respective stage were as follows:Straight-line driving. The distance between the starting point and the end point of the vehicle was about 300 m. There was no obvious bend between the starting point and the end point, and the vehicle could reach the end point without making large turns. There were no other traffic participants in this scene.Curved driving. The distance between the starting point and the end point of the vehicle was about 500 m. There was an obvious bend between the starting point and the end point. The vehicle needed to make a turn to reach the end point. There were no other traffic participants in this scene.Navigation in dynamic scenes. The distance between the starting point and the end point of the vehicle was about 700 m. The starting point and the end point were not set artificially, and the starting point and the end point were randomly generated. There were other traffic participants in this scene, and the autonomous driving car participated in the traffic flow together with them.

2.Vehicle and sensor installation

The experimental vehicle for autonomous driving used the Audi A6 2010_blue model in VTD, which can control the vehicle’s steering wheel, throttle, and brake. The input range of the direction wheel was [−1, 1] (0 represented the horizontal value of the direction wheel), and the input range of the accelerator and brake was [0, 1] (0 represented the relaxed state, and 1 represented the full state). The installed sensors in the vehicle included an RGB camera, a GNSS sensor, and an IMU sensor. The installation location of the on-board GNSS sensor was (x = 1.0, y = 0, z = 2.8). Table 3 lists the parameters, installation position, and imaging of the on-board RGB camera.

3.Weather conditions

To fully consider the effect of weather factors on decision making and planning, following the research experience of Xiaodan Liang et al. [23] from Carnegie Mellon University, this study divided weather into two groups. One included sunny days, clear sunsets, daytime rain, and post-rain days, while the other included cloudy days and rainy sunsets. The specific situation is shown in Figure 9.

VTD is capable of simulating real-time high-quality lighting and shadow effects, road reflections, vehicle rendering, weather effects, sensor image rendering, headlight visual effects, and third-person-perspective image generation, with the aim of creating realistic traffic scenarios. In general, weather adjustments in VTD comprise support for time of day (morning, noon, afternoon, or night), lighting conditions, ground wetness, clouds, rain, snow, fog, and other customizable settings. In this study, VTD’s SCP commands were adopted to randomly control the weather conditions and light sources. For instance, the following command set a cloudy weather condition at noon:


*<Environment><Friction value=“1.000000” /><Date day=“1” month=“6” year=“2008” /><TimeOfDay headlights=“auto” timezone=“0” value=“42400” /><Sky cloudState=“off” visibility=“100000.000000” /><Precipitation intensity=“0.000000” type=“none” /><Road effectScale=“0.500000” state=“dry” /></Environment>*


4.Network training parameters

To be specific, due to the limitation of GPU memory size, the batch size was set to 128, and the learning rate started at 0.001. If the difference in the average accuracy determined in the validation process two consecutive times was within 1%, the learning rate would be multiplied by 0.1. The model input was set to 4 × 224 × 224 and was normalized with the specific mean and variance. The maximum number of epochs was set to 200. In the experiment, Adam served as an optimizer to update parameters, and the cross-entropy function served as the loss function. In the above-described experiment, a polynomial learning rate decay strategy called “Poly” was employed; its equation is expressed below:(22)$$lr\times (1-\frac{iteration}{\max\_iteration})\mathrm{power}$$
where *lr* denotes the initial learning rate (set to 0.01 for this experiment), power expresses the decay rate (set to 0.9), max_*iteration* represents the maximum number of iterations for training, and *iteration* is the current training iteration.

In terms of the task, this study focused on testing and verifying the performance of the vehicle in completing decision-making and planning tasks under straight-line driving, turning, and dynamic traffic with the goals of traffic efficiency and driving safety in mind. Thus, the agent did not consider the impact of factors such as traffic lights and yellow lines during training to enhance its adaptability to dynamic environments.

Due to the different data ranges of the respective sensor, the data range of each sensor was limited and normalized to facilitate data processing, with the range of data being restricted to [−1, +1]. For some sensors whose data ranges were infinite, such as the distance of the vehicle from the centerline of the track, although the data range was infinite, in reality, when the data range exceeds [−1, +1], the vehicle had already driven off the track. Accordingly, we did not normalize this type of data. The data range of the controller also was limited. For steering, +1 represented a maximum left turn, and −1 represented a maximum right turn. In terms of braking, 0 represented no braking, and 1 expressed full braking. For acceleration, 0 represented the minimum throttle with zero acceleration, and +1 denoted the maximum throttle with maximum acceleration.

To avoid meaningless training processes for the vehicle, it was defined that if any of the following situations occurred during the experiment, the episode of the experiment would end. This processing made the training process of the vehicle more efficient. Table 4 lists the definitions of the end conditions.

After 48 h of training the agent on the Germany_2018 map, a total of 10,000 steps were trained. With the rise in the number of training iterations, the accumulated reward obtained by the agent tended to be increased and then stabilized. The curve in Figure 10 represents the change in the accumulated reward obtained through the agent during training at each iteration. The blue line represents the smoothed curve of the average cumulative reward with a window size of 100.

As depicted in Figure 10, at around 1000 steps, the agent had already mastered a suitable driving strategy through the interaction with the environment and continued to improve its driving strategy in later learning processes. Thus, the effectiveness of the interaction between the agent and the environment was verified.

#### 3.2.2. Performance Metrics

In the early research on semantic segmentation, segmentation accuracy was a key metric of interest. In this study, the mean Intersection over Union (*mIoU*) ratio served as the metric to measure the segmentation accuracy. The calculation process was as follows:(23)$$mIoU=\frac{1}{k+1}\sum_{i=0}^{k}\frac{p_{ii}}{\sum_{i=0}^{k}p_{ij}+\sum_{i=0}^{k}p_{ji}-p_{ii}}$$
where *k* is the number of categories; *k* + 1 is the number of categories with the background introduced; *p_ij_* is to predict category *i* as category *j*, which is equivalent to *FN*; *p_ji_* is to predict category *j* as category *i*, which is equivalent to *FP*; and *p_ii_* is to predict category *i* as category *i*, which is equivalent to *FP*.

To validate the effectiveness of the proposed auxiliary segmentation task module, we conducted testing and verification based on different simulated routes in VTD. The results were quantitatively evaluated using *mIoU* (Table 5). The IoU for each category of each trained model was evaluated based on its own dataset’s test set. Additionally, Figure 11a,b show the epochs, cross-entropy loss, and *mIoU* for the four trained models. The x-axis in the loss curve represents the training epochs, while the y-axis represents the value of the loss function. Due to the significant fluctuations in the loss function values, we applied smoothing to the function for better visualization. The curve in the graph is the smoothed function.

We first conducted ablation experiments on the validation path to adjust the balance of losses. Table 6 shows the results with and without the use of auxiliary tasks in the presence of obstacles. We can see that by adding auxiliary tasks on the valuation path, the agent’s performance was improved. The agent’s average speed was closer to the target speed, its position was closer to the center of the road, and the deviation from its trajectory was smaller. Additionally, the agent traveled a longer average distance per episode, and the overall variance of the measurements was lower.

The performance of the trained models was examined in two different time periods (i.e., with illumination and without illumination) to fully validate the effectiveness of the proposed algorithm and the generalization performance of the trained models. For the ease of the experimental result analysis, the model trained using image features was termed RGB, while the model trained using image and depth features was termed RGB-D. Four navigation tasks (i.e., straight driving, single-turn driving, navigation without dynamic obstacles, and navigation with dynamic vehicles and pedestrians) were examined 25 times in daytime and complex nighttime scenes. Table 7 lists the results of the average task-completion rates.

In terms of scenarios with temporal variations to nighttime or weather variations to rainy conditions, certain biases existed in the prediction results. As indicated by the analysis results, the generalization of weather conditions notably affected the pixel data. For instance, under nighttime and rainy conditions, the pixel value data were directly affected such that the model’s judgment of object boundaries and confidence was affected. Accordingly, the overall data turned out to be more challenging to identify, suggesting that the samples generated through weather generalization had successfully built extreme scenario cases that reached or approached the lower limits of model perception and prediction, which was conducive to verifying the robustness and reliability of the model.

As depicted in Table 7, for the midday testing, the trained model in this study had already learned the lane-keeping strategy to a large extent in the absence of vehicles and pedestrians. Moreover, the incorporation of depth features led to a significant improvement in the average task completion rate. During the nighttime testing, due to domain gaps, the model trained on daytime scenes did not transfer well to nighttime scenes. The reason for the above result was that nighttime scenes display poor illumination, resulting in changes in image data compared with daytime conditions. In addition, the depth-estimation method was dependent on the assumption of photometric consistency, which was partially violated in dim nighttime environments. However, compared with the model trained solely on image features, the model integrating image and depth features exhibited poor performance in tasks involving turns and both dynamic and non-dynamic obstacles during nighttime testing. As revealed by the above result, the model trained solely on image features was more sensitive to domain gaps, whereas the inclusion of depth features mitigated the degradation arising from domain gaps to a certain extent.

Subsequently, tests were performed on the completion of the three driving tasks that were designed before the agent was installed, and the results were quantitatively and qualitatively analyzed. In the experiment, the sensor information from the VTD driving simulation served as input. In several tasks, additional moving obstacles were generated, and Gaussian noise was introduced to the environmental perception detection results to simulate modeling errors in a real environment.

To demonstrate the effectiveness of the decision-making and planning algorithm proposed in this study in handling multiple obstacles and different weather conditions, the DDPG model proposed in this study was compared with three existing benchmark models in a fair comparison. The three benchmark models were the conventional modular framework (MP), imitation learning (IL), and reinforcement learning (RL) [24], and the fair comparisons were performed in three tasks of gradually increasing difficulty: driving in a straight line, turning on bends, and dynamic navigation. To be specific, the baseline MP decomposed the autonomous driving task into three subsystems (i.e., perception, planning, and control). The local planning implemented a rule-based preset strategy completely dependent on the perception module’s awareness of the environment. The baseline IL took input from front camera images and steering control commands and directly trained the model using human driving videos through super-vised learning. Furthermore, the baseline RL was a deep reinforcement learning frame-work employing the Asynchronous Advantage Actor–Critic (A3C) [9] algorithm.

The respective task was initiated at a randomly selected starting point in a sample similar to the one presented in Figure 11, and each group of tasks was examined more than 25 times. The quantitative analysis mainly focused on the statistical analysis of the completion of different decision-making and planning model tasks. Table 5 shows the percentage of successful completion of tasks under each scenario for the examined conventional modular framework (MP), imitation learning (IL), reinforcement learning (RL), and our proposed decision-making and planning model that incorporated dynamic scene information (Ours). The higher the percentage of successful completion, the better the performance of the model. The quantitative equation was as follows:(24)$$Arrive=100\times \left(\frac{count\_achieve}{count\_all}\right)$$
where count_achieve represented the number of times the agent successfully completed the task, and count_all represented the total number of tests performed on the agent to complete the task.

It can be observed in Table 8 that the decision-making and planning model proposed in this study (Ours) was generally superior to all the above baseline methods under all conditions and was especially better than the reinforcement learning (RL) baseline model. In addition, the model proposed in this study had a high degree of robustness in dynamic environments, and although the test results were not perfect, it performed better than other methods. For instance, the completion rate of 60% in dynamic navigation achieved by our model represented a significant improvement over the IL model and RL model, which achieved completion rates of only 24% and 2%, respectively.

On the other hand, the performance of the original reinforcement learning (RL) model was significantly worse than all other methods, even after training for 12 days in the simulator. The reason for the above result was that RL-based autonomous driving systems are fragile [25] and require very time-consuming exploration to obtain reasonable results. Unlike video games such as Atari [26] and maze navigation [27], practical tasks such as autonomous driving require complex decisions. Relying solely on visual information can lead to serious problems with sample inefficiency and parameter-search failure.

During a test process, the agent made a short left turn before entering a straight road but encountered another vehicle ahead of it on the lane. When the decision-making agent determined that the surrounding environment was safe for a lane change, the autonomous driving vehicle executed a lane-change maneuver and moved into the adjacent lane. After completing the overtaking action, the road started to curve to the right. The following records show the changes in the vehicle’s steering angle and speed during this test.

The time interval between the respective step was 0.1 s, and it can be seen in Figure 12a,b that the agent began to overtake at around step 500. At this time, a significant change existed in the steering angle to complete the overtaking maneuver, and the vehicle speed was also increased notably after completing the overtaking maneuver.

## 4. Discussion

There were many cases of failure in the training process, mainly due to the inappropriate setting of the reward function, which caused the algorithm to become stuck in some local optimal points. Figure 13 depicts a failure case in lane keeping in which it can be seen that the policy remained near a reward value of 0.2 until the end. The major reason for the above result was the absence of a suitable penalty reward for the vehicle leaving the road. If the vehicle crossed a lane line that it should not have crossed, it would receive a negative reward value. This led the network to believe that it would not receive negative reward values later if it left the road directly at the beginning, and from another perspective, this meant obtaining a larger cumulative reward. Thus, the network decided to end the round by leaving the road directly in each episode.

The above-mentioned failed training cases also fully demonstrated a problem with current reinforcement learning methods: the difficulty of setting the reward function. The design of an excellent reward function should guide the algorithm to implement the expected functions and consider the learning effect of the algorithm, some possible loopholes, and some local optimal solutions that are easy to fall into. At present, the practical reward function mainly relies on human experts to summarize based on experimental experience. If the design of the reward function can also be automated, the performance and applicability of reinforcement learning methods will be further enhanced.

## 5. Conclusions

In this study, a novel method was proposed to address the challenges facing deep reinforcement learning (DRL) techniques in the development of autonomous driving systems. To be specific, a multi-task DDPG decision-making strategy was proposed that integrated a DRL-based decision-making main task with an image semantic segmentation-based auxiliary task for enhancing the generalization ability of the agent. Moreover, a joint simulation platform was developed and implemented based on VTD–CarSim and TensorFlow to expedite the testing and verification of the DRL algorithms. As indicated by the experimental results, the proposed method achieved a minimum completion rate of 60% and a maximum completion rate of 96% in a wide variety of driving tasks under different weather conditions. In general, this study can facilitate the in-depth development of DRL-based autonomous driving technology and lay a solid basis for subsequent research in the relevant field.

## Figures and Tables

**Figure 1 sensors-23-07021-f001:**
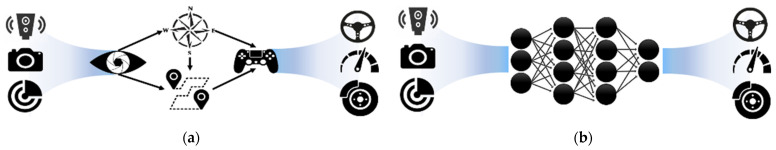
General intelligent vehicle system framework diagram: (**a**) model-based framework; (**b**) end-to-end framework.

**Figure 2 sensors-23-07021-f002:**
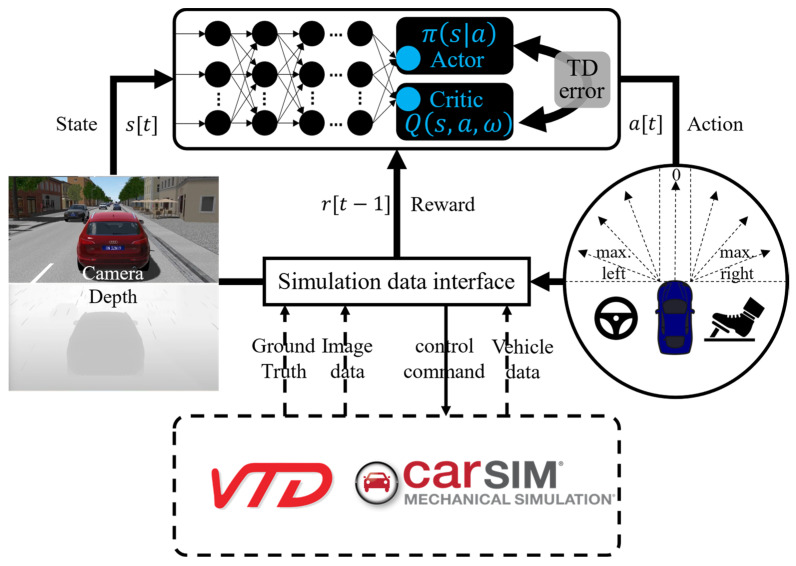
Framework of the motion control system.

**Figure 3 sensors-23-07021-f003:**
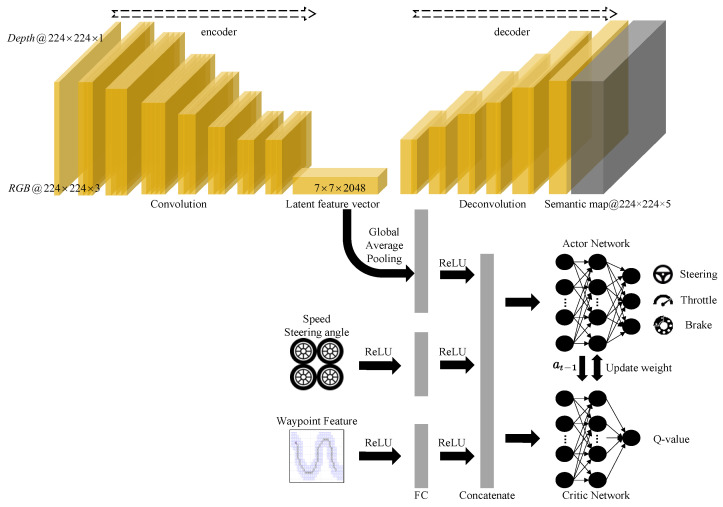
Reinforcement learning architecture with semantic segmentation auxiliary task.

**Figure 4 sensors-23-07021-f004:**
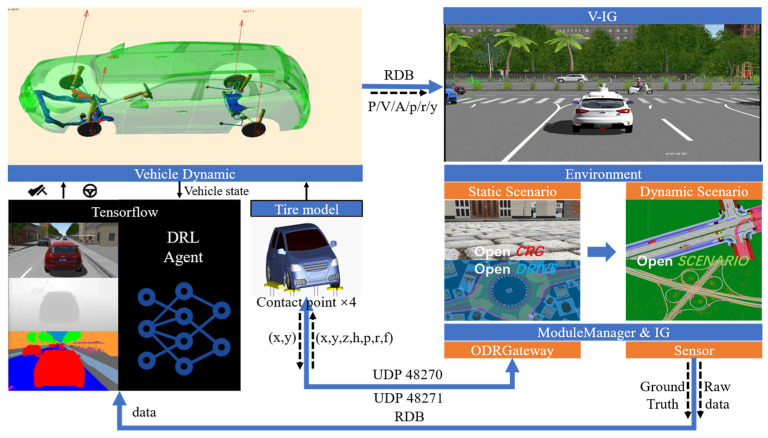
Simulation environment data flow diagram. P represents position, V represents velocity, A represents acceleration, p represents the pitch angle, r represents the roll angle, and y represents the yaw angle.

**Figure 5 sensors-23-07021-f005:**
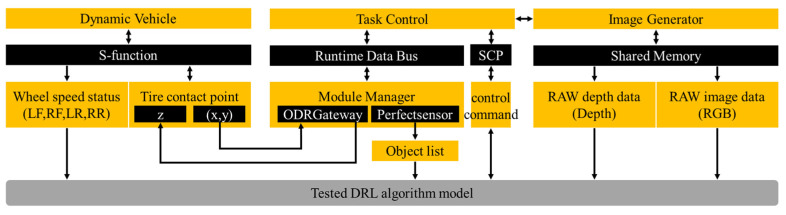
Data flow diagram of the VTD–CarSim–TensorFlow–Simulink interface.

**Figure 6 sensors-23-07021-f006:**
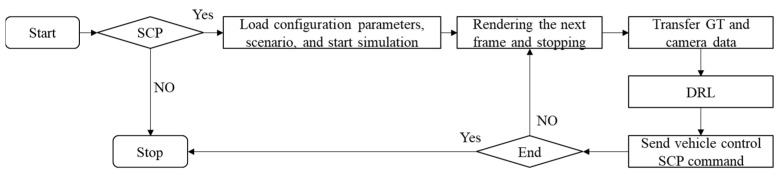
Simulation control flow diagram of the integrated toolchain.

**Figure 7 sensors-23-07021-f007:**
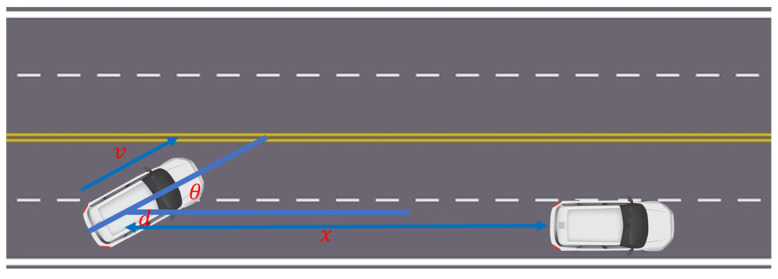
Symbols used in the reward function.

**Figure 8 sensors-23-07021-f008:**
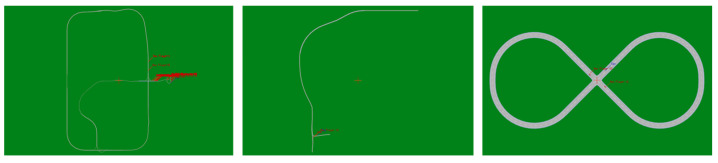
Germany_2018 map in VTD.

**Figure 9 sensors-23-07021-f009:**
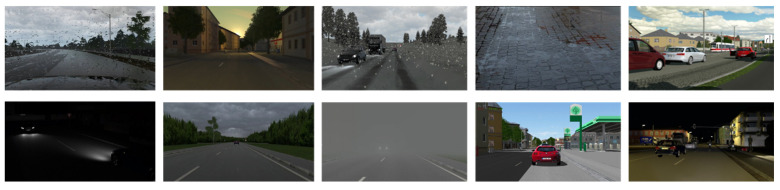
Effect of weather on images.

**Figure 10 sensors-23-07021-f010:**
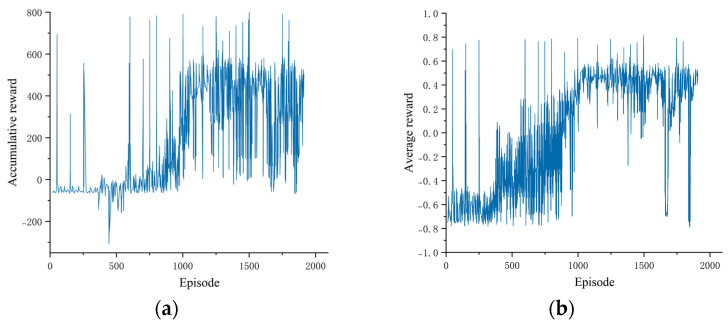
Cumulative reward change curve: (**a**) cumulative reward curve; (**b**) average reward curve.

**Figure 11 sensors-23-07021-f011:**
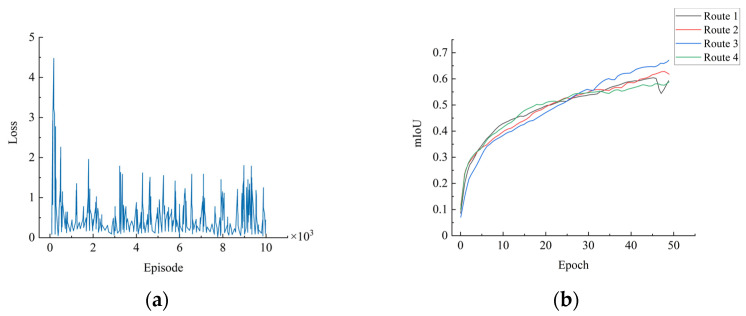
Test result curves for semantic segmentation assistance: (**a**) loss curves of the trained models; (**b**) *mIoU* curves for different routes.

**Figure 12 sensors-23-07021-f012:**
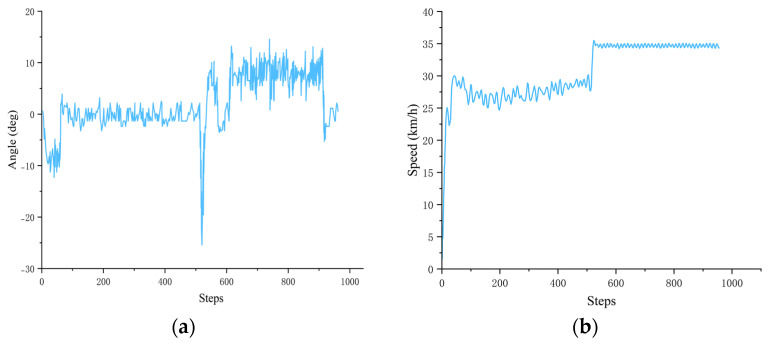
(**a**) Record of the steering angle of the autonomous driving agent during this dynamic navigation task. (**b**) Record of the vehicle speed of the autonomous driving agent during this dynamic navigation task.

**Figure 13 sensors-23-07021-f013:**
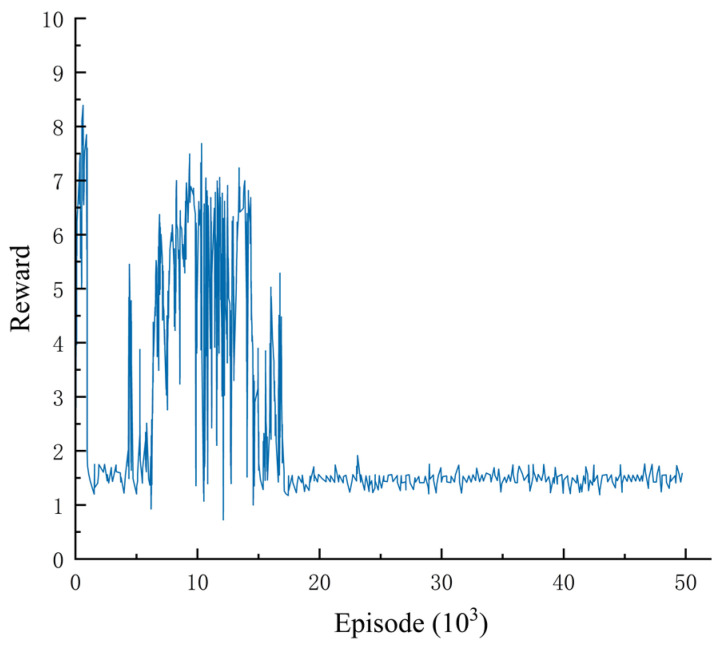
Failure case in lane keeping.

**Table 1 sensors-23-07021-t001:** Parameters of CarSim vehicle dynamics model.

Parameter	Mark	Value
Sprung mass/kg	*m*	1650
Tread/m	*d*	1.36
Wheelbase/m	*L*	3.05
Distance from center of mass to front axle/m	*a*	1.4
Distance from center of mass to rear axle/m	*b*	1.65
Yaw inertia/kg·m^2^	*Iz*	3234
Front axle cornering stiffness/(N/rad)	*Cf*	−62,618
Rear axle cornering stiffness/(N/rad)	*Cr*	−110,185
Vehicle speed/(km/h)	*u*	90
Road adhesion coefficient	*μ*	0.4

**Table 2 sensors-23-07021-t002:** Introduction of software and hardware dependencies.

Software/Hardware Dependency	Version
CPU	Intel Core i7-9750H 2.60 GHz
GPU	NVIDIA GeForce RTX 3090 24 G
Memory	Kingston DDR4 16 G
Software editing environment	PyCharm Community Edition 2020.1
Main programming language	Python 3.7

**Table 3 sensors-23-07021-t003:** Camera parameters and images.

Camera Parameter Name	Value
FOV (fovH,fovV)	(120, 80)
Installation location (x,y,z)	(2.0052, 0, 1.1025)
Installation angle (H,P,R)	(0, 0, 0)
Intrinsic matrix (fx,fy,cx,cy,k1,k2,k3,p1,p2)	(1934.393162, 1898.933992, 1901.148339, 1041.813981, −0.273666977882385, 0.051446333527565, 0.0, −0.000620494422037154, 0.00188898132182658)
Effective pixels (width,height)	(1920, 1080)
RGBD image and semantic image	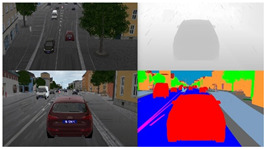

**Table 4 sensors-23-07021-t004:** Episode-end condition definition.

Episode-End Condition Definition:
1. Reaching the maximum preset step;
2. The vehicle ran out of the track or collided with the edge of the track;
3. The vehicle drove in reverse;
4. The speed of the vehicle was extremely low.

**Table 5 sensors-23-07021-t005:** *mIoU* testing metrics for different routes.

Class	Route 1	Route 2	Route 3	Route 4
Road	0.8960	0.9751	0.9219	0.9062
Sidewalk	0.6966	0.7838	0.3903	0.6810
Traffic sign	0.1595	0.0000	0.0771	0.1439
Vehicle	0.6914	0.4971	0.4601	0.5780
*mIoU*	0.7497	0.4708	0.4288	0.7207

**Table 6 sensors-23-07021-t006:** Results of our baseline training on our benchmark.

	Average Speed during One Episode (km/h)	Average Cross-Track Error (m)	Average Angle between Car and Road (°)	Traveled Distance in One Episode (m)	The Number of Episodes that Ended with a Collision
Without auxiliary task	28.7	2.5	25	358.2	24/36
With auxiliary task	35.4	1.8	7.5	547.3	25/36

**Table 7 sensors-23-07021-t007:** Test results of average completion with sun and without sun.

Task Types	With Sun		Without Sun	
	RGB	RGBD	RGB	RGBD
Straight driving	94.45	95.90	64.78	82.32
Single-turn driving	92.01	95.73	14.00	50.67
Navigation without dynamic obstacles	88.12	98.34	2.00	10.34
Navigation with dynamic vehicles and Pedestrians	73.03	90.00	3.90	10.01

**Table 8 sensors-23-07021-t008:** Comparison of task-completion rates with other autonomous driving decision-making models.

Task Type	Weather	MP	IL	RL	Ours
Straight driving	With sun	92	97	74	96
Straight driving	Without sun	50	80	68	82
Cornering	With sun	61	59	12	74
Cornering	Without sun	50	48	20	78
Dynamic navigation	With sun	24	38	2	60
Dynamic navigation	Without sun	44	42	4	64

## Data Availability

The dataset is available upon request from the corresponding author.

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
