# Peer review of "A Multi-Task Fusion Strategy-Based Decision-Making and Planning Method for Autonomous Driving Vehicles"

_sensors, 2023, doi:10.3390/s23167021_

Round 1

Reviewer 1 Report

This paper presents a novel method to address the challenges faced by deep reinforcement learning techniques in the development of autonomous driving systems.

A more detailed exposition is lacking in some of the sections discussed, but the article covers all the topics associated with the object of study. More research work based on semantic segmentation is also lacking. The bibliography should be updated, including more recent works.

It would be advisable to explain the aspects related to semantic segmentation. Recently Meta has published the paper "Introducing Segment Anything: Working toward the first foundation model for image segmentation". It would be interesting to compare the proposed system with this one, from the point of view of its capabilities and possible application in the scope of the work done.

The authors should explain in more detail what is the contribution of the CarSim simulation package to the proposed system.

The influence of weather conditions on the results is not very clear. The weather conditions under which the experiments were performed should be indicated in more detail.

Some comments:

In Figure 10 and 16 the resolution should be improved.

No comments.

Author Response

Please refer to the Word attachment for the response to the reviewer’s comments.

Reviewer 2 Report

The paper presents timely topic with certain novelty. Before possible publication, the following comments should be considered:

1- The contributions should be carefully highlighted in the introduction section. 

2- The introduction can be enriched with recent papers in the realm of autonomous vehicles such as "distributed finite time neural network observer based consensus tracking control of heterogeneous underwater vehicles "

3- Why author used an RL algorithm?

4- The conclusion section should be supported by some data. 

Proof reading is needed. 

Author Response

(The authors gave the same response as above.)

Reviewer 3 Report

Regarding the paper, I think the author has done an interesting study. The originality of his work is clear and focuses on two points:

-          - Development of autonomous driving system and coupling it to the reinforced Learning program.

-         - The reward function was considered dynamic instead of static, which improved the result.

This paper examines autonomous driving technology based on Deep Reinforcement Learning (DRL) and proposes an innovative method for addressing the challenges it faces. The study focuses on improving the generalization capability of the autonomous driving agent using a DDPG decision-making and planning method based on a multi-task fusion strategy. The article points out that the development of real vehicles is costly, time-consuming and risky. To speed up the testing and verification of DRL algorithms, a simulation platform based on VTD+Carsim+Tensorflow+Simulink has been designed and implemented. This platform enables faster, more efficient testing, while providing a realistic simulation environment for training and validating autonomous driving models. Experimental results show that the proposed method is effective in improving the performance and generalizability of the vehicle control algorithm.

However, it should be noted that in the absence of further information in the paper, it is difficult to give a precise explanation of the use of the specific loss function in the DRL techniques described in this paper. A more detailed explanation or reference to other work would be necessary to fully understand the rationale behind the use of cross-entropy as a loss function in this particular context.

In addition, the paper mentions experimental results, but does not provide details of the algorithm's performance during training and predictions. For a full evaluation of the algorithm, it would be necessary to know the performance measures used, the input data and the parameters used to assess the algorithm's accuracy and generalizability.

Author Response

(The authors gave the same response as above.)
